# Comment on Marra et al. Metastatic Renal Cell Carcinoma to the Soft Tissue 27 Years after Radical Nephrectomy: A Case Report. *Medicina* 2023, *59*, 150

**DOI:** 10.3390/medicina59020409

**Published:** 2023-02-20

**Authors:** Pietro De Luca, Luca de Campora, Angelo Camaioni

**Affiliations:** Head and Neck Department, San Giovanni-Addolorata Hospital, 00184 Rome, Italy

Recently, Marra et al. [1] published a case report in which they commented on their experience in the management of a 73-year-old woman who presented with a left temporal bone metastasis from renal cell carcinoma (RCC) 27 years after the upfront treatment (January 1997, right radical nephrectomy). We congratulate the authors for the interesting article, but due to our largest experience being in the management of head and neck metastases from other districts’ tumors, we want to discuss the correct step-by-step management.

The authors wrote that “most malignant tumor metastases are reported within 5 years of the original diagnosis of the primary tumor. Nevertheless, ‘late metastases’ do happen”. According to the literature, several studies reported that late recurrence >5 years after resection is a known biologic behavior of RCC [2,3].

In the case report, the authors wrote, “In November 2021, the patient underwent surgery to remove the lesion. The massive bleeding and the texture of the lesion did not allow the removal”; these difficulties with homeostasis during biopsy and the presence of bleeding at the first visit were reported by Nishii et al. [4] in a large cohort of patients. RCC has a significant place among studies of angiogenesis due to its high incidence rate and dense vascularity [5]; according to these findings and the report of previous bleeding during the biopsy, we strongly believe that preoperative vascular embolization should be performed in order to prevent massive hemorrhage during biopsy and surgery for pulsatile oral metastasis of RCC, even in late-stage oncological patients.

Furthermore, the authors reported that the patient is actually treated with Pazopanib 200 mg (1 pill per day and 2 pills of 200 mg per day, on alternate days), a protein kinase inhibitor; targeted therapies, such as tyrosine kinase inhibitors and mammalian target of rapamycin inhibitors have replaced traditional chemotherapy regimens in the treatment of metastatic renal cell carcinoma, despite their toxicity being still debated [6,7]. However, a large number of patients enrolled in these clinical trials experienced lung or pancreatic metastases, while a small cohort of patients with metastases in the head and neck was included.

According to our experience, in a patient with a solitary, accessible metastatic lesion, a surgical approach is standard practice, eventually followed by CH or target therapy, but also the combination of TKI with immune checkpoint inhibitors to shrink tumors and decrease blood flow, followed by surgery, are also sufficient options.

In the last 20 years (January 2003–January 2023), at the Head and Neck Dept. of San Giovanni-Addolorata Hospital (Rome, Italy), we surgically treated 16 patients with metastases from RCC in the head and neck region. Twelve patients were males (12/16, 75%), and four patients were females (4/16, 25%); the mean age was 71.1 years (range 57–85; SD 9.55). The sites involved were the tongue (5/16, 31%), scalp, oral cavity, the skin of the nose (each site with 3 patients, 3/16, 18.7%), and thyroid (2/16, 12.5%); 8 patients developed lung metastases (8/16, 50%).

The duration from the onset of renal cancer to head and neck metastasis ranged from 3 to 17 years (SD 3.93), with an average of 4.8 years; among them, 8 patients (8/16, 50%) showed metastases after more than 10 years after upfront treatment, while other 6 patients (6/16, 37.5%) showed relapse of the disease in a time between 5 and 10 years after surgery; only 2 patients showed metastases in head and neck area in less than 5 years.

Follow-up data were available for all the patients; the mean follow-up was 10.4 years (range 4–19 years; SD 4.32); at the last follow-up, 4 patients died from the metastatic spread of the disease (4/16, 25%), 4 were alive with evidence of the disease (4/16, 25%), 6 patients were alive with no evidence of disease (6/16, 37.5%), while 2 patients died for other causes (ictus cerebri, heart failure; 2/16, 12.5%). 

Our data suggested that despite late metastases being rare, patients with RCC should be proposed for a longer follow-up time, and surgical management of metastases in the head and neck region should be treated in a tertiary center after a proper multidisciplinary step-by-step evaluation.

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
