# Peer review of "Comment on Marra et al. Metastatic Renal Cell Carcinoma to the Soft Tissue 27 Years after Radical Nephrectomy: A Case Report. Medicina 2023, 59, 150"

_medicina, 2023, doi:10.3390/medicina59020409_

Round 1

Reviewer 1 Report

What the authors comment to the manuscript "Metastatic Renal Cell Carcinoma to the Soft Tissue 27 Years after Radical Nephrectomy: A Case Report" is quite correct. The description is  almost the same as what I thought, and I appreciate you for describing this paper.

I agree with the authors that surgery with vascular embolization should be the first choice. However, the authors only discuss surgical therapy as the only option. The tone of their argument is as if they are denying drug therapy. I think it is not good. Tyrosine kinase inhibitors (TKI) such as pazopanib, which the authors of the original paper used, and the current combination of TKI with immune checkpoint inhibitors to shrink tumors and decrease blood flow, followed by surgery, are also sufficient options. I think I should add that.

The patient's tumor was difficult to remove due to significant bleeding and should be reoperated with TAE, but shrinking the tumor and decreasing blood flow with TKI or ICI+TKI followed by surgery is also an option. I think I should add that.

Author Response

Dear Reviewer, thanks for your suggestion, which we truly appreciated. We added the therapeutic strategy you suggested in the paper.

Thanks for your work.

Best regards